# Allicin May Promote Reversal of T-Cell Dysfunction in Periodontitis via the *PD-1* Pathway

**DOI:** 10.3390/ijms22179162

**Published:** 2021-08-25

**Authors:** Shankargouda Patil, Mohammed E. Sayed, Maryam H. Mugri, Khalaf F. Alsharif, Arif Salman, Shilpa Bhandi, Hosam Ali Baeshen, Thodur Madapusi Balaji, Pradeep Kumar Yadalam, Saranya Varadarajan, R. Srimathi R. Radha, Kamran Habib Awan, Vikrant R. Patil, A. Thirumal Raj

**Affiliations:** 1Department of Maxillofacial Surgery and Diagnostic Sciences, Division of Oral Pathology, College of Dentistry, Jazan University, Jazan 45412, Saudi Arabia; 2Department of Prosthetic Dental Sciences, College of Dentistry, Jazan University, Jazan 45412, Saudi Arabia; drsayed203@gmail.com; 3Department of Maxillofacial Surgery and Diagnostic Sciences, College of Dentistry, Jazan University, Jazan 45412, Saudi Arabia; dr.mugri@gmail.com; 4Department of Clinical Laboratory Sciences, College of Applied Medical Sciences, Taif University, P.O. Box 11099, Taif 21944, Saudi Arabia; alsharif@tu.edu.sa; 5Department of Periodontics, School of Dentistry, West Virginia University, P.O. Box 9490, Morgantown, WV 26506, USA; salman.abdulshakore@hsc.wvu.edu; 6Department of Restorative Dental Sciences, College of Dentistry, Jazan University, Jazan 45412, Saudi Arabia; shilpa.bhandi@gmail.com; 7Department of Orthodontics, College of Dentistry, King Abdulaziz University, Jeddah 21589, Saudi Arabia; drbaeshen@me.com; 8Tagore Dental College and Hospital, Chennai 600127, India; tmbala81@gmail.com; 9Department of Periodontics, Adhi Parashakthi Dental College, Melvaruvathur 603319, India; drpradeepkumar@apdch.edu.in; 10Department of Oral Pathology and Microbiology, Sri Venkateswara Dental College and Hospital, Chennai 600130, India; vsaranya87@gmail.com (S.V.); thirumalraj666@gmail.com (A.T.R.); 11SRM College of Pharmacy, Kancheepuram 603203, India; srimathr@srmist.edu.in; 12College of Dental Medicine, Roseman University of Health Sciences, South Jordan, UT 84095, USA; kamranhabibawan@gmail.com; 13Biogenre Private Limited, Pune 412105, India; patilvikrant.r@gmail.com

**Keywords:** allicin, gingival crevicular fluid, periodontitis, T lymphocytes

## Abstract

We evaluated the role of allicin in periodontitis using an in silico and in vitro design. An in silico docking analysis was performed to assess the plausible interactions between allicin and *PD-L1*. The cytokine profile of gingival crevicular fluid (GCF) samples obtained from periodontitis patients was estimated by cytometric bead array. CD3+ lymphocytes isolated from the peripheral blood were sorted and characterized using immunomagnetic techniques. Cultured and expanded lymphocytes were treated with the GCF samples to induce T-cell exhaustion. Optimum concentrations of allicin were added to exhausted lymphocytes to compare the expression of *TIM-3* and *LAG-3* gene expression at baseline and post-treatment. Allicin was found to bind to the *PD-L1* molecule as revealed by the in-silico experiment, which is possibly an inhibitory interaction although not proven. GCF from periodontitis patients had significantly higher concentrations of TNF-α, CCL2, IL-6, IFN-γ, and CXCL8 than controls. GCF treatment of CD3+ lymphocytes from the periodontitis patients significantly increased expression of T-cell exhaustion markers *TIM-3* and *LAG-3*. Allicin administration with GCF treatment resulted in significant lowering of the expression of exhaustion markers. Allicin may exert an immunostimulatory role and reverse immune-destructive mechanisms such as T-cell exhaustion.

## 1. Introduction

Periodontitis is a disease with global prevalence. It affects the gingival tissues and attachment apparatus of the teeth. Periodontitis manifests in the formation of periodontal pockets and abscesses. It leads to episodic pain and tooth mobility [1]. The disease is initiated and perpetuated by gram-negative putative periodontal pathogens colonizing the dental plaque biofilm. Periodontitis is a risk factor for systemic diseases such as type 2 diabetes mellitus [2], cardiovascular disease [3], and adverse pregnancy outcomes [4]. 

A substantive body of research has investigated the pathogenesis of periodontitis to develop effective therapeutic strategies to cure the condition. Earlier research revealed the immunological characteristics of periodontitis lesions. A microbial challenge mounted by putative periodontal pathogens such as *Porphyromonas gingivalis*, *Tannerella forsythia* and *Treponema denticola* in the dental plaque biofilm causes the movement of leukocytes into the gingival tissues to combat the infection. The initial inflammation in the gingiva is characterized by the abundant presence of polymorphonuclear leukocytes or neutrophils [5]. They are slowly replaced by the lymphocytes with time [5]. Both T and B lymphocytes participate in periodontitis pathogenesis. T cells exist in various subtypes denoted as CD4+ helper T cells, CD8+ killer T cells, and regulatory T cells that help to regulate periodontal homeostasis. Helper T cells exist in various subpopulations termed Th0, Th1, Th2, Th17, and Treg, which produce different cytokines [6]. These cells recognize antigens processed by antigen-presenting cells. They help in mediating humoral responses by stimulating the proliferation and expansion of B-cells. Killer T cells, also known as large granular lymphocytes, exert direct cytotoxic actions on the microbial invaders. Lymphocyte-mediated responses are double-edged swords in the sense that they protect the host but at the same time cause periodontal tissue destruction. T cells that produce elevated levels of interleukin-17 are primarily responsible for bone loss in periodontitis [7]. Clinical studies have demonstrated the presence of interleukin-17 in gingival crevicular fluid and gingival tissue homogenates of periodontitis patients [8]. As the disease progresses, there is a shift from the cells that do not secrete immunoglobulins to immunoglobulin-secreting cells evidenced by B-cell activation. IgG and IgM are the major immunoglobulin subtypes secreted by the lesion resident cells. Elevated levels and activation of B-cells lead to bone loss as the disease progresses [9].

T-cell exhaustion is a phenomenon that occurs in chronic conditions such as periodontal disease and cancer. A state of T-cell dysfunction occurs after chronic exposure of T cells to antigens [10]. Classically, T-cell exhaustion was demonstrated to occur in cytomegalovirus infections in the mouse model [11]. Exhaustion is characterized by reduced proliferation, responsiveness, self-renewal, cytotoxicity, and cytokine production by T cells. This leads to peripheral immune tolerance, and the infection is not completely resolved [12]. The cardinal cellular signs of T-cell exhaustion include upregulation of inhibitory receptors such as programmed cell death (*PD-1*), T-cell immunoglobulin and mucin-domain containing TIM-3, and lymphocyte-activation gene (*LAG-3*). This can cause inhibition of the cell signaling pathways that activate T cells [12]. Earlier research has shown that the *PD-1* pathway is upregulated in the T cells of mice with chronic cytomegalovirus infection [11]. PD-1 is a member of the CD 28 superfamily of molecules and is a putative marker of T-cell exhaustion [13]. The *PD-1* pathway is regulated by the binding of *PD-1* to the *PD-L1* ligand. The interaction of programmed cell death ligand -1(*PD-L1*) with *PD-1* triggers downstream signaling pathways that leads to T-cell exhaustion [14]. Other markers such as *LAG-3* and *TIM-3* also are coexpressed and upregulated in T-cell exhaustion along with the activated PD-1 pathway. *LAG-3* acts synergistically with *PD-1*, causing severe T-cell exhaustion. Inhibition of *LAG-3* and *TIM-3* markers quickly revives exhaustion of the affected T cells [15]. 

In periodontitis, the cytokine milieu is possibly the main factor implicated in T-cell exhaustion. An increase in the counts of periodontal pathogens such as *Porphyromonas gingivalis* could also account for the T-cell exhaustion phenomenon due to upregulation of *PD-L1*. Previous research indicated that *Porphyromonas gingivalis* infection of oral squamous cell carcinoma cell lines causes a significant *PD-L1* increase [16]. A study on periapical lesions demonstrated upregulated *PD-1* and *LAG-3* in the lesion environment [17]. A similar upregulation of *PD-1* has been reported in samples obtained from patients with periodontitis [18]. Plant-derived metabolites may provide a therapeutic strategy for reviving T-cell exhaustion in periodontitis through the *PD-1* pathway checkpoints. Earlier research has shown three herb-derived molecules that show promise—curcumin from *Curcuma longa*, baicalin from the Chinese herb *Scutellaria baicalensis*, and the triterpenoid saponin platycodin-D from the roots of *Platycodon grandiflorum*. These herbs have traditionally been used in Chinese herbal medicine [19]. These herbs may exert beneficial effects in periodontitis through inhibition of *PD-L1*. Garlic possesses similar immunomodulatory properties and is far easier to source. The chemical allicin from garlic exerts immunostimulant activities [20]. The immunostimulatory actions of allicin have been elaborated through experiments that have demonstrated increased humoral immune response manifested as elevated immunoglobulin levels after administration of allicin [21]. It was also demonstrated that allicin upregulates cytokine production and also stimulates cell mediated immune response, especially with regard to lymphocytes [22]. Allicin may affect the upregulation of the *PD-1* pathway. This exploratory study was designed to examine whether allicin could inhibit *PD-L1* at the in-silico level. We investigated the effects of allicin administration on T-cell exhaustion and revival. We also evaluated whether gingival crevicular fluid obtained from patients with periodontitis could cause dysfunction of T cells obtained from the same patients in an in vitro culture setting. 

## 2. Results

An in-silico design was used to study the binding interaction between allicin and *PD-L1*. The single-dimensional structure of allicin is depicted in Figure 1A. Figure 1B depicts the docked configuration of allicin and *PD-L1*. We observed that allicin interacts with PD-L1 through hydrogen bonds and hydrophobic interactions. Table 1 shows the binding energy (kcal/mol) for docked complex, demonstrating the interaction between allicin and *PD-L1* in various positional confirmations. At position 7, a negative binding energy of −7.10 kcal/mol could be elicited. This depicts a very high affinity and interaction energy between allicin and *PD-L1*. We concluded from the in-silico data that allicin could possibly be an inhibitor of *PD-L1*.

The present study had an in vitro component using GCF samples collected from five patients with periodontitis. These samples were used for cytokine profile analysis and addition to T-lymphocyte cultures for induction of T-cell dysfunction. The lymphocyte cultures that underwent GCF treatment were also expanded from CD3+ lymphocytes separated from blood samples obtained from the patients with periodontitis. Five systemically healthy subjects with healthy periodontal status were chosen as the control. GCF samples were obtained from the control for comparison of cytokine profile with the periodontally diseased GCF samples. The results of the study are presented in further sections.

### 2.1. GCF from Periodontitis Patients Show Higher Levels of Proinflammatory Cytokines Compared to Healthy Controls

A total of 13 cytokines were evaluated and compared, including IL-4, IL-2, CXCL10, IL-1β, TNF-α, CCL2, IL-17A, IL-6, IL-10, IFN-γ, IL-12p70, CXCL8, and TGF-β1 (Figure 2A–M). TNF-α, CCL2, IL-6, IFN-γ, and CXCL8 showed significantly higher levels in GCF from periodontitis patients compared to control subjects (*p* < 0.05) (Figure 2E,F,H,J,L, Table 2). 

### 2.2. MTT Assay for Evaluation of Cytotoxic Concentrations of Allicin

Sorted CD3+ T lymphocytes obtained from the peripheral blood samples were maintained in culture and expanded. The cells were treated with different concentrations of allicin and incubated for 48 h (Figure 3A–J). The MTT assay showed that higher concentrations of allicin (100 and 250 μM) were toxic to T cells. Cellular viability was found to be decreased at the higher concentrations of allicin. Lower concentrations of allicin (0.5, 1, and 2.5 μM) increased the metabolic activity of T cells. No significant change was observed in the viability of T cells with optimal concentrations of allicin at 5, 10, 25, and 50 μM (Figure 3K, Table 3).

### 2.3. Treatment of CD3+ T Cells with GCF from Periodontitis Increases the Gene Expression of TIM-3 and LAG-3

The cultured CD3+ T cells were activated in the presence of various concentrations of GCF from periodontitis. After treatment, the cells were incubated for periods between 24 and 120 h. The gene expression of *TIM-3* and *LAG-3* increased significantly in CD3+ T cells in a dose-dependent and time-dependent manner (*p* < 0.05). The highest level of gene expression was in presence of 2 μL/mL of GCF after 48–72 h of the incubation (Figure 4A–E and Figure 5A–E, Table 4 and Table 5). Hence, this concentration was chosen for further experiments.

### 2.4. Allicin Treatment Decreased GCF-Induced Upregulation of TIM-3 and LAG-3

Treatment with 2 μL/mL GCF of CD3+ T cells induced the highest levels of *TIM-3* and *LAG-3* gene expression. Hence, the same concentration of GCF was used to analyze the effect of allicin. T cells were incubated with 2 μL/mL of GCF and simultaneously treated with different concentrations of allicin for 24–120 h. Allicin treatment decreased the GCF-induced *TIM-3* and *LAG-3* expression in T cells in a dose-dependent and time-dependent manner. The effect was very evident at a concentration of 25 μM between the 48 and 96 h incubation period (*p* < 0.05) (Figure 6A–E and Figure 7A–E, Table 6 and Table 7).

## 3. Discussion

The present study explored the effect of allicin on T-cell dysfunction and activation. We sought to elucidate the possible host modulatory effects of allicin derived from garlic in the management of periodontitis. The choice of CD3+ lymphocytes was based on the fact that CD3+ cells depict a population helper and cytotoxic T cells, which are very critical in periodontitis pathogenesis. Moreover, the Th response is also largely dependent on T cells. It has been proven beyond doubt that antibody production by B cells is dependent on T-cell responses. Hence, we assessed the effect of allicin on the above cells

Activation of the *PD-1* pathway can result in T-cell exhaustion, characterized by T-cell dormancy and reduced immune response. For the in-silico analysis, a molecular docking study was performed to examine the interaction between the allicin and *PD-L1* pathway. The results revealed that allicin is bound to *PD-L1* with high affinity and could possibly cause inhibition of *PD-L1* although not demonstrated in the present study. If further proven in an in vivo environment, this would translate to a blockade of the *PD-1* checkpoint pathway and prevention of T-cell exhaustion. From this result, it can be extrapolated that allicin could possibly be a potent phytochemical inhibitor of T-cell exhaustion. 

We conducted a wet lab assay to explore in vitro induction and reversal of T-cell dysfunction in periodontitis. GCF samples and CD3+ T cells were obtained from patients with periodontitis. T cells were expanded, cultured, and treated with the GCF samples of the periodontitis patients, and T-cell dysfunction was assessed. CD3+ T lymphocytes were obtained from the patients following immunosorting. CD3+ is a specific cell surface marker and a coreceptor of both helper T cells and cytotoxic T cells. GCF is an exudate that is produced in excess amounts during the pathogenesis of periodontal inflammation [23]. It is an innate host defense mechanism and contains a large number of host-derived protective molecules [24]. GCF isolated from periodontitis patients is a cocktail of inflammatory cytokines. In periodontitis, the GCF contains large amounts of cytokines and proinflammatory mediators such as IL-1 beta [25] and TNF alpha [26] that are responsible for periodontal destruction. We found that the GCF samples obtained from periodontitis patients contained significantly higher proportions of cytokines such as TNF-α, CCL2, IL-6, IFN-γ, and CXCL8 compared to the samples obtained from systemically and periodontally healthy controls. There were also detectable levels of other inflammatory mediators such as TGF-β1, IL-1β, IL-2, IL-4, IL-10, IL-12p70, and IL-17A in the GCF samples of periodontitis patients. The effects of the GCF treatment on CD3 + T cells varied according to the concentration and treatment duration. There was a dose- and time-dependent elevation in the expression of two genes, *TIM3* and *LAG3*, in T cells following GCF treatment. These findings suggest that T-cell exposure to the GCF led to T-cell dysfunction.

T-cell exhaustion is a cellular and molecular mechanism that occurs in many chronic and viral conditions. T-cell exhaustion is characterized by hierarchical loss cytotoxicity, mitosis, and IL-2 production, followed by the production of IFN-γ and TNF-α. Researchers have described T-cell exhaustion as a peripheral tolerance mechanism where T cells lose their phenotypic functions. During the exhaustion phenomenon, there is a characteristic upregulation of inhibitory receptors on T cells. A study of LCMV-infected mice revealed the significant role of inhibitory receptors following a blockade of the programmed death 1 (PD-1) pathway. PD-1 and its ligand play a pivotal role in this regard [27]. *PD-1* is an inhibitory receptor subtype of the CD28 superfamily. It is upregulated on exhausted CD8 T cells during chronic LCMV infection. A blockade of this pathway can rejuvenate and restore CD8 T-cell function and enhance the resolution of the viral infection. *PD-1* seems to act as a checkpoint molecule in T-cell exhaustion. *TIM-3* and *LAG-3* are considered markers of T-cell exhaustion. These surface molecules are coexpressed on exhausted T cells along with *PD-1* [28]. These markers act synergistically to cause and maintain T-cell exhaustion. We found *TIM-3* and *LAG-3* to be positively overexpressed in T cells following GCF treatment. 

A blockade of *PD-1*-, *TIM-3*-, and *LAG-3*-mediated cell signaling pathways can reverse T-cell exhaustion. Therapeutic strategies that modulate the *PD-1* checkpoint have been proposed. Herbal extracts and herb-derived active molecules such as platycodin-D, baicalin, and curcumin are thought to modulate the *PD-1* pathway and revive T-cell activity. Allicin (diallylthiosulfinate) is a defense molecule derived from the herb garlic. In our study, T cells were treated with the GCF cytokine cocktail and allicin. We observed that allicin restored T-cell function significantly by downregulating the expression of *TIM-3* and *LAG-3* genes. Allicin had a significant effect on reviving T-cell dysfunction at a concentration of 25 μM after 48–96 h of incubation. This concentration was not cytotoxic to T cells as evidenced by the MTT cytotoxicity assay.

*Allium Sativum* L, commonly referred to as garlic, exhibits immunomodulatory properties. On tissue damage, the enzyme alliinase catalyzes the conversion of the amino acid alliin to the thiosulfinate allicin [29]. Decomposition of allicin results in the formation of bioactive molecules such as diallyl disulfide (DADS), diallyl sulfide (DAS), dithiins, and ajoene [30]. Allicin exerts antibacterial, anti-inflammatory, antioxidant, and immunostimulant activities in a rabbit model of Pasteurella infection [31]. Earlier research demonstrated that allicin can modulate T cells and adhesion molecules, inhibit NF-*κ*B activation, and thereby prevent liver damage [32]. Previously, in vitro models reported that allicin exhibits anti-inflammatory activity, inhibits the secretion of TNF-*α* secretion, and protects intestinal epithelial cells [33].

Allicin is easily synthesized as its natural source is abundant. Allicin may have applications as a host modulatory strategy in periodontal therapy where inflammation, a diminished immune response, dysregulated oxidant–antioxidant mechanisms, and T-cell exhaustion have a role to play. Allicin could be a candidate for addition to mouthwashes, toothpaste, and intrapocket delivery systems to exert topical effects. Further studies on animal models of ligature-induced periodontitis may help develop a fuller picture of allicin’s ability to reverse T-cell exhaustion. Research in this regard can unearth further applications of allicin as a perioceutical agent in the management of destructive periodontal disease.

## 4. Materials and Methods

Scientific research (IRB), College of Dentistry, Jazan University approved the study (Reference no.CODJU-19233). The study was comprised of an in silico component performed by a trained team of bioinformatic specialists, while the in vitro studies were carried out in the molecular biology laboratory. 

### 4.1. Protocol and Methodology of the In Silico Experiment

The crystal structure of the programmed cell death ligand 1(PD-L1) intended as the receptor binding to the allicin ligand was used in the docking study (5NIU). The protein was processed by eliminating certain specific ligands. PyRx virtual screening software (Vetsio 9.0, La Jolla, California, United States of America) was used to prepare for docking. The protein was further processed into a PDBQT file format to facilitate readability in the AutoDock software (Vetsio 9.0, La Jolla, California, United States of America). The three-dimensional configuration of allicin was extracted from the PUBCHEM database. The ligand was prepared for the docking process by the addition of charges and optimization using the universal force field. The ligand structure was converted into the PDBQT file format to facilitate readability in the AutoDock software. The one-dimensional structure of the allicin ligand is depicted in Figure 1A. Molecular docking was performed using AutoDock Tools (ADT) version 1.5.6 and AutoDock version 4.2.6 docking program [34]. Gasteiger charges were applied during docking calculations. Molecular docking of the RBD domain of PD-L1 with the allicin ligand was carried out. Minimization of obtained best pose was carried out using the UCSF Chimera program (version 1.15, San Francisco, United States of America). Minimization was carried out using the Amber14SB force field for protein and GAFF force field for drug molecules. Visualization and analysis of the binding mode as well as interactions in the binding pocket of the obtained poses were analyzed using the PYMOL, PLIP [35]. The results were analyzed and interpreted.

### 4.2. Patient Selection 

Written informed consent was obtained from all patients who participated in the study. Five patients who were diagnosed with periodontitis after intraoral examination and periodontal charting were enrolled in the study. The diagnosis of periodontitis was made according to the 2018 World Workshop classification of periodontal diseases, which is the most updated classification system available [36]. An additional five study participants who were periodontally healthy with the absence of systemic disease were selected to be the control group. The control group was used to compare the cytokine profile of GCF samples before experimentation and cell culture studies. The exclusion criteria for the study were the presence of systemic diseases, history of periodontal treatment, intake of antibiotics, antioxidants, analgesics, and immunomodulatory drugs in the last 6 months. Patients who were taking any herbal supplements were also excluded from the study. Smokers and pregnant and lactating mothers were also excluded from the study.

### 4.3. Gingival Crevicular Fluid and Blood Sample Collection

After a thorough clinical examination, the patients were seated upright in the dental chair with good illumination. GCF was sampled from the deepest periodontal pocket in each quadrant of the patient’s mouth using microcapillary pipettes using a standard protocol [37]. Isolation before sample collection was performed using cotton rolls and high-volume suction to prevent saliva contamination. GCF samples contaminated with blood were discarded and not used for further analysis. The GCF collected from the 4 sites in each patient was pooled together and centrifuged at 3000× *g* and stored at −80 degrees centigrade until further analysis. The same sampling protocol was followed for the systemically and periodontally healthy subjects to obtain pooled GCF samples from 4 periodontally healthy sites. The GCF samples obtained from the control subjects were used for cytokine analysis to compare the levels of cytokines in the GCF samples of periodontitis patients. No further analysis was conducted on the samples from the control subjects. One portion of the sampled GCF from each periodontitis patient was used to analyze the cytokine profile, while the other portion was used for in vitro lymphocyte treatment to induce T-cell dysfunction. A 5 mL blood sample was collected from each periodontitis patient from the antecubital vein using a sterile aseptic protocol. The blood sample was immediately transported to the molecular biology facility for isolation of CD3 + cell isolation and culture of the cells. The GCF sample collected from a periodontitis patient was used to induce T-cell dysfunction in the T cells obtained and cultured from the same patient to simulate a clinical scenario in an in vitro setting.

### 4.4. Cytometric Bead Array for the Estimation of Cytokine Levels in the GCF Samples

To determine the levels of the cytokines in the GCF samples, a cytometric bead array was performed. The LEGENDplex^TM^ human essential immune response panel (13-plex) (BioLegend, San Diego, CA, USA) was used for cytokine detection (TNF-alpha, TGF-β1, IL-1β, IL-2, IL-4, IL-6, IL-10, IL-12p70, IL-17A, IFN-γ, CCL2, CXCL8, and CXCL10). The experimental protocol followed the manufacturer’s instructions. In total, 25 μL of the GCF samples was incubated for 2 h with the microbeads. The detection antibodies were added to the tests after incubation and incubated for 30 min. Following incubation, the samples were washed with the wash buffer. The samples were then centrifuged at 2000 rpm for 5 min. Following supernatant elimination, the pellet was resuspended in a 200 μL sheath fluid. The samples were acquired using a flow cytometer (Attune NxT, Thermo Fisher Science, Waltham, MA, USA) and analyzed using LEGENDplex^TM^ data analysis software (BioLegend, San Diego, CA, USA).

### 4.5. Magnetic Cell Sorting of CD3+ T Cells and T-Cell Culture

Total CD3+ T lymphocytes were isolated with magnetic cell sorting using a human-specific CD3+ microbead kit (Miltenyi Biotec, Bergisch Gladbach, Germany) from restored peripheral blood mononuclear cells (PBMCs). Sorted CD3+ T cells were subject to culture using TexMACS immune cell-culturing medium (Miltenyi Biotec, Bergisch Gladbach, Germany). Activation and expansion was induced by using human T-cell activation/expansion kit (Miltenyi Biotec, Bergisch Gladbach, Germany) with addition of 100 U/mL of IL-2 (Miltenyi Biotec, Bergisch Gladbach, Germany).

### 4.6. MTT Assay for Analysis of Allicin Cytotoxicity

Cell viability was measured using an MTT assay. The cells were seeded into 96-well plates (Nunc, Rochester, NY, USA) at a density of 5 × 10^3^ cells per well and incubated for 24 h. The adhered cells were treated with complete medium (DMEM + 10% FBS) mixed with allicin (Santa Cruz Biotechnology, Santa Cruz, CA, USA) of various concentrations ranging between 0.5 and 250 µM. The samples were incubated for 48 h. After incubation, MTT solution (0.5 mg/mL) (Sigma-Aldrich Corp., St. Louis, MO, USA) was added to each well and incubated for four hours at 37 °C. Postincubation, the medium was removed, and 100 µL dimethyl sulfoxide (DMSO) (Sigma-Aldrich, St. Louis, MO, USA) was added to each well. Multiskan FC spectrophotometer (Thermo Scientific, San Jose, CA, USA) was used to measure the absorbance at 570 nm.

### 4.7. Treatment of T Cells with GCF and Allicin

CD3+ T cells were cultured in a 24-well plate (Nunc, Rochester, NY, USA) with a density of approximately 1 million cells/well. Cells were treated with different volumes of GCF (1, 2, and 5 µL/mL) and were incubated for 24, 48, 72, 96, and 120 h. Similarly, T cells were treated with different concentrations (5, 10, 25, and 50 µM) of allicin along with 2 µL/mL GCF for the same time points. T cells without GCF treatment were regarded as controls.

### 4.8. RT-PCR for Gene Expression Analysis

The total RNA was extracted from treated T cells with GeneJet purification columns (Invitrogen, Thermo Scientific, Vilnius, Lithuania). High-capacity cDNA reverse-transcription kit (High Capacity, Applied Biosystems, Carlsbad, CA, USA) was used for conversion of 1 μg of total RNA into cDNA. Quantitative analyses of gene expressions were conducted by using the SYBRGreen PCR master mix (Applied Biosystems, Austin, TX, USA) on QuantStudio 5 real-time PCR system (Applied Biosystems, Foster City, CA, USA). Respective primers (IDT, Coralville, IA, USA) were used for the analysis. The primer sequence of the housekeeping GAPDH gene, TIM-3, and LAG-3 are presented in Table 8. Expressions of the target genes TIM-3 and LAG-3 were normalized to GAPDH, using the ΔCt technique. 

### 4.9. Analysis of the Data and Statistics

The experiments were performed in triplicates and the obtained results are presented as mean ± standard deviation. Each experimental group was individually compared with the control group, and the data were analyzed using an unpaired t-test (two-tailed) on GraphPad Prism 8 software (GraphPad Software, La Jolla, CA, USA). The level of significance was set at a *p*-value of less than 0.05, and a *p*-value less than 0.01 was interpreted as highly significant.

## 5. Conclusions

This study set out to examine whether allicin derived from the herb garlic could reverse T-cell dysfunction. In our study, allicin bound and inhibited PD-L1 in the in silico model. Allicin administration alleviated T-cell dysfunction by downregulating the TIM-3 and LAG-3 genes in a dose-dependent and time-dependent manner. Allicin may have beneficial biological effects by reversing T-cell dysfunction in chronic conditions. It may have clinical applications in treating T-cell-mediated diseases and chronic inflammatory disorders.

## Figures and Tables

**Figure 1 ijms-22-09162-f001:**
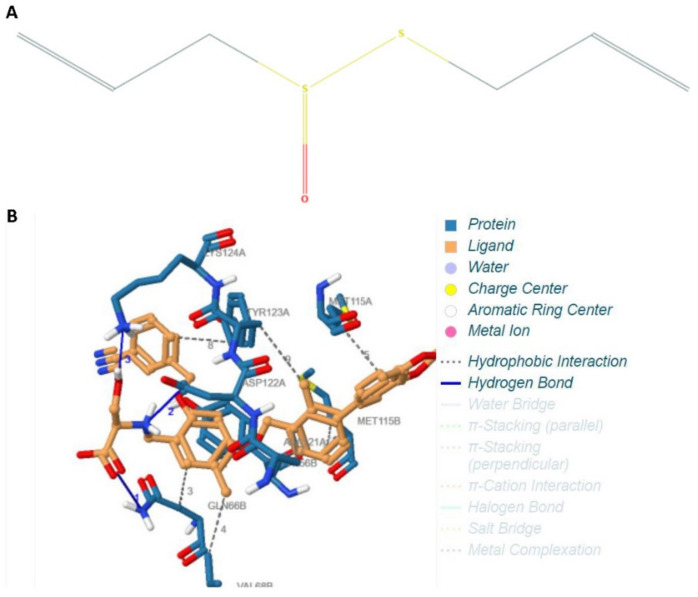
Structure and docking of allicin. (**A**) One-dimensional chemical structure of allicin. (**B**) Docked configuration of allicin and *PD-L1*.

**Figure 2 ijms-22-09162-f002:**
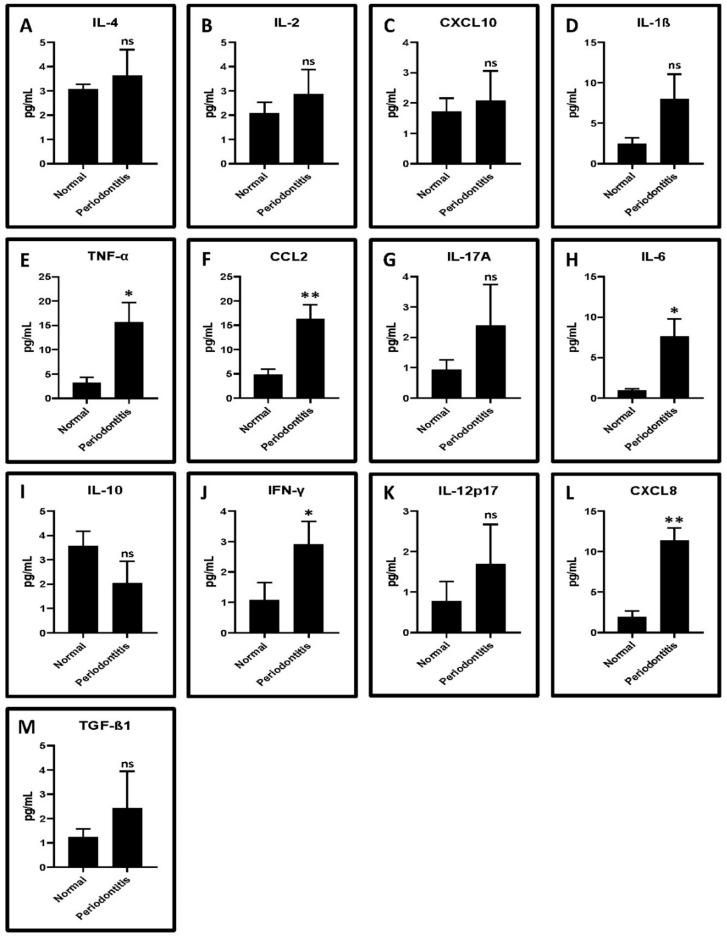
Cytokine levels in periodontally healthy versus diseased GCF samples. (**A**–**M**) Flow-cytometry-based analysis of human cytokines. The quantitation and comparative analysis of cytokines in the GCF samples were assessed by a cytometric bead array on the flow cytometer. ns not significant, * *p* < 0.05, ** *p* < 0.01. IL-4: interleukin 4, IL-2: interleukin 2, CXCL10: C-X-C motif chemokine ligand 10, IL-1β: interleukin 1 beta, TNF-α: tumor necrosis factor alpha, CCL2: C-C motif chemokine ligand 2, IL-17A: interleukin 17A, IL-6: interleukin 6, IL-10: interleukin 10, IFN-γ: interferon gamma, IL-12p70: interleukin 12, CXCL8: interleukin 8, and TGF-β1: transforming growth factor beta 1.

**Figure 3 ijms-22-09162-f003:**
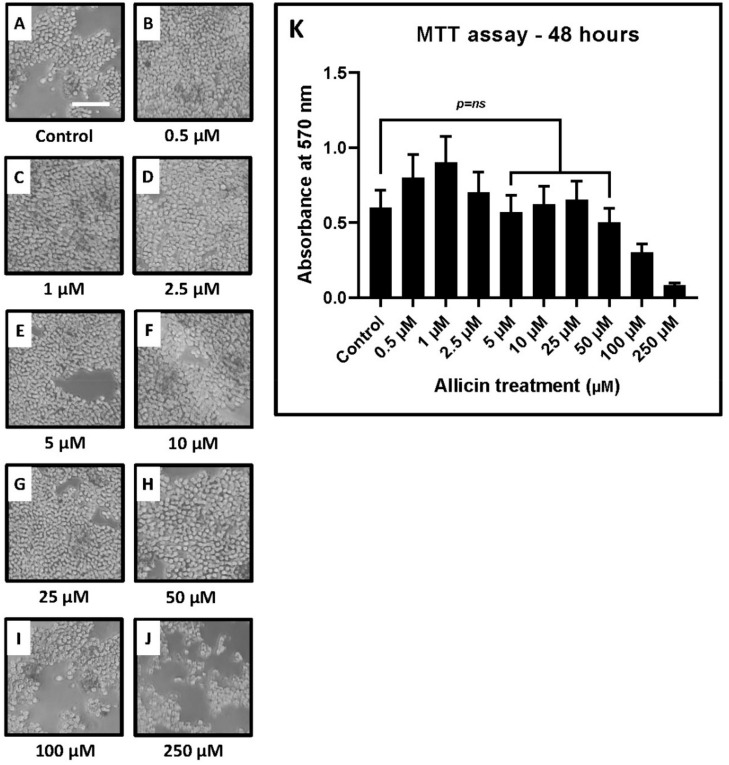
MTT assay depicting the influence of various concentrations of allicin on human CD3+ lymphocytes. Treatment of sorted CD3+ T lymphocytes with different concentrations of allicin and MTT assay for assessment of the viability of the treated cells. (**A**–**J**) Micrographs of CD3+ T lymphocytes treated with various concentrations of allicin (0.5, 1, 2.5, 5, 10, 25, 50, 100, and 250 μM). Scale bar = 100 µm. (**K**) Comparative cell viability in CD3+ T lymphocytes treated with various concentrations of allicin.

**Figure 4 ijms-22-09162-f004:**
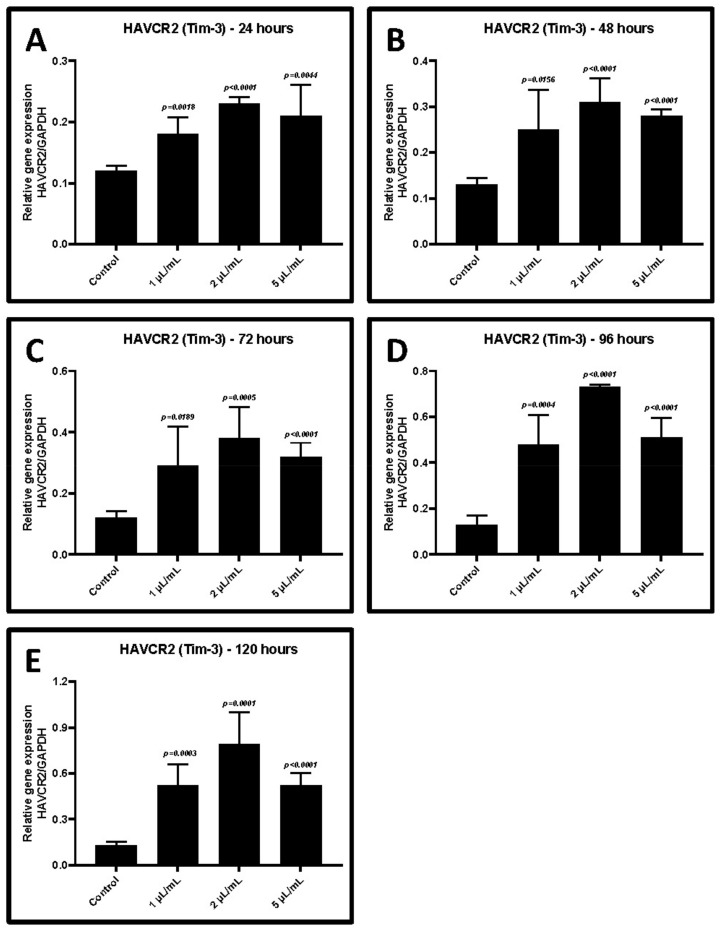
Relative gene expression of *HAVCR2 (TIM-3)* in CD3+ T lymphocytes treated with different concentrations of GCF (1, 2, and 5 μL/mL). (**A**) Comparative gene expression of *TIM-3* after 24 h of treatment. (**B**) Comparative gene expression of *TIM-3* after 48 h of treatment. (**C**) Comparative gene expression of *TIM-3* after 72 h of treatment. (**D**) Comparative gene expression of *TIM-3* after 96 h of treatment. (**E**) Comparative gene expression of *TIM-3* after 120 h of treatment. *HAVCR2 (TIM-3)*: T-cell immunoglobulin and mucin-domain containing-3, GCF: Gingival crevicular fluid.

**Figure 5 ijms-22-09162-f005:**
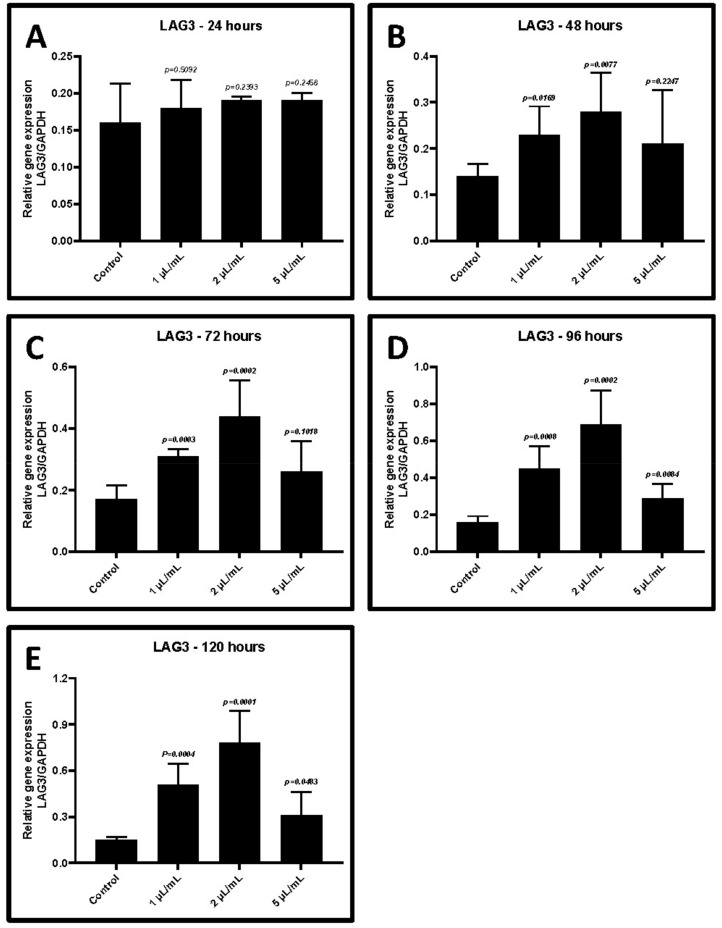
Relative gene expression of *LAG-3* in CD3+ T lymphocytes treated with different concentrations of GCF (1, 2, and 5 μL/mL). (**A**) Comparative gene expression of *LAG-3* after 24 h of treatment. (**B**) Comparative gene expression of *LAG-3* after 48 h of treatment. (**C**) Comparative gene expression of *LAG-3* after 72 h of treatment. (**D**) Comparative gene expression of *LAG-3* after 96 h of treatment. (**E**) Comparative gene expression of *LAG-3* after 120 h of treatment. *LAG-3*: lymphocyte-activation gene 3, GCF: gingival crevicular fluid.

**Figure 6 ijms-22-09162-f006:**
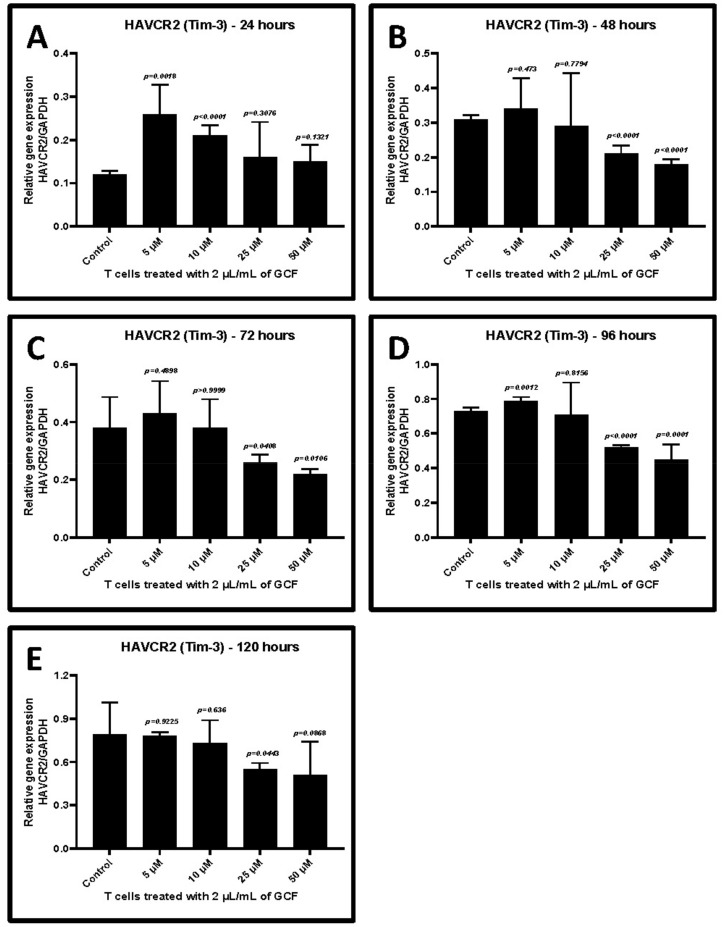
Relative gene expression of *HAVCR2 (TIM-3)* in 2 μL/mL GCF-treated CD3+ T lymphocytes treated with different concentrations of allicin (5, 10, 25, and 50 μM). (**A**) Comparative gene expression of *TIM-3* after 24 h of treatment. (**B**) Comparative gene expression of *TIM-3* after 48 h of treatment. (**C**) Comparative gene expression of *TIM-3* after 72 h of treatment. (**D**) Comparative gene expression of *TIM-3* after 96 h of treatment. (**E**) Comparative gene expression of *TIM-3* after 120 h of treatment. *HAVCR2 (TIM-3)*: T-cell immunoglobulin and mucin-domain containing-3, GCF: gingival crevicular fluid.

**Figure 7 ijms-22-09162-f007:**
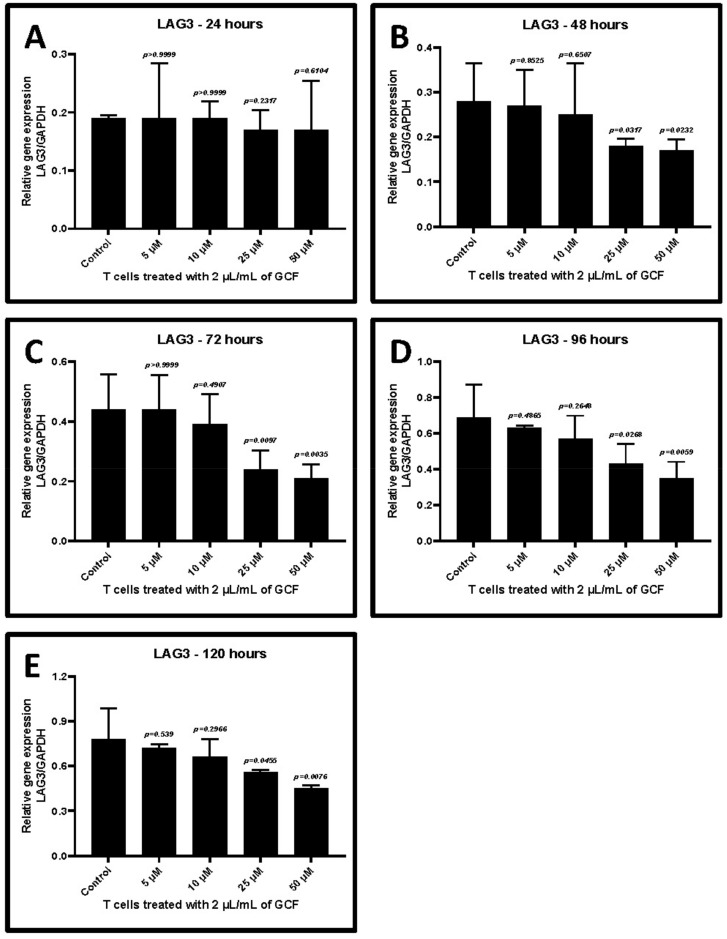
Relative gene expression of *LAG-3* in 2 μL/mL GCF-treated CD3+ T lymphocytes treated with different concentrations of allicin (5, 10, 25, and 50 μM). (**A**) Comparative gene expression of *LAG-3* after 24 h of treatment. (**B**) Comparative gene expression of *LAG-3* after 48 h of treatment. (**C**) Comparative gene expression of *LAG-3* after 72 h of treatment. (**D**) Comparative gene expression of *LAG-3* after 96 h of treatment. (**E**) Comparative gene expression of *LAG-3* after 120 h of treatment. *LAG-3*: lymphocyte-activation gene 3, GCF: gingival crevicular fluid.

**Table 1 ijms-22-09162-t001:** Binding energy (kcal/mol) for docked complex demonstrating the interaction between allicin and *PD-L1*.

Rank Pttern	Sub—Rank	Run	Binding Energy	Cluster Rmsd	Reference Rmsd
1 RANKING	1	7	−7.10	0.00	118.43
1 RANKING	2	3	−7.10	0.67	118.33
1 RANKING	3	10	−7.10	0.21	118.40
1 RANKING	4	6	−7.02	0.88	118.31
2 RANKING	1	8	−6.84	0.00	129.46
3 RANKING	1	1	−6.70	0.00	129.39
3 RANKING	2	4	−6.63	0.63	129.38
4 RANKING	1	9	−5.88	0.00	130.20
5 RANKING	1	2	−5.56	0.00	122.91
6 RANKING	1	5	−5.27	0.00	118.23

**Table 2 ijms-22-09162-t002:** Analysis of cytokines in the GCF from periodontitis patients (Mean ± SD).

Cytokines	Normal	Periodontitis	*p*-Value
IL-4	3.06 ± 0.21	3.64 ± 1.06	0.4999
IL-2	2.09 ± 0.44	2.86 ± 1.03	0.37
CXCL10	1.71 ± 0.45	2.08 ± 0.97	0.6387
IL-1β	2.49 ± 0.73	7.97 ± 3.08	0.065
TNF-α	3.24 ± 1.06	15.63 ± 4.08	0.0101 *
CCL2	4.88 ± 1.04	16.29 ± 2.92	0.0036 *
IL-17A	0.93 ± 0.34	2.39 ± 1.34	0.2094
IL-6	0.97 ± 0.18	7.59 ± 2.19	0.0101 *
IL-10	3.57 ± 0.59	2.034 ± 0.9	0.0831
IFN-γ	1.08 ± 0.57	2.90 ± 0.76	0.0311 *
IL-12p17	0.78 ± 0.48	1.69 ± 0.97	0.2755
CXCL8	1.92 ± 0.76	11.32 ± 1.58	0.0006 *
TGF-β1	1.25 ± 0.33	2.43 ± 1.51	0.3474

* *p* < 0.05.

**Table 3 ijms-22-09162-t003:** MTT assay for cell viability of CD3+ T lymphocytes treated with different concentrations of allicin.

Treatment	OD at 48 h of Allicin Treatment (Mean ± SD)	*p*-Value
Control w/o treatment	0.62 ± 0.12	-
0.5 μM allicin	0.78 ± 0.15	0.0488
1 μM allicin	0.86 ± 0.17	0.0386
2.5 μM allicin	0.73 ± 0.14	0.0479
5 μM allicin	0.57 ± 0.11	0.7625
10 μM allicin	0.62 ± 0.12	0.8462
25 μM allicin	0.65 ± 0.13	0.6405
50 μM allicin	0.50 ± 0.09	0.3169
100 μM allicin	0.34 ± 0.06	0.0162
250 μM allicin	0.08 ± 0.02	0.0016

**Table 4 ijms-22-09162-t004:** Relative gene expression of *TIM-3* in CD3+ T lymphocytes treated with different concentrations of GCF (Mean ± SD).

GCF Treatment	Control w/o Treatment	1 μL/mL	2 μL/mL	5 μL/mL
24 h	0.12 ± 0.01	0.18 ± 0.03	0.23 ± 0.01	0.21 ± 0.05
*p*-value	-	0.0018	<0.0001	0.0044
48 h	0.13 ± 0.01	0.25 ± 0.09	0.31 ± 0.05	0.28 ± 0.01
*p*-value	-	0.0156	<0.0001	<0.0001
72 h	0.12 ± 0.02	0.29 ± 0.13	0.38 ± 0.10	0.32 ± 0.04
*p*-value	-	0.0189	0.0005	<0.0001
96 h	0.13 ± 0.04	0.48 ± 0.13	0.73 ± 0.01	0.51 ± 0.08
*p*-value	-	0.0004	<0.0001	<0.0001
120 h	0.13 ± 0.02	0.52 ± 0.14	0.79 ± 0.21	0.52 ± 0.08
*p*-value	-	0.0003	0.0001	<0.0001

**Table 5 ijms-22-09162-t005:** Relative gene expression of *LAG-3* in CD3+ T lymphocytes treated with different concentrations of GCF (Mean ± SD).

GCF Treatment	Control w/o Treatment	1 μL/mL	2 μL/mL	5 μL/mL
24 h	0.16 ± 0.05	0.18 ± 0.04	0.19 ± 0.001	0.19 ± 0.01
*p*-value	-	0.5092	0.2393	0.2453
48 h	0.14 ± 0.03	0.23 ± 0.06	0.28 ± 0.08	0.21 ± 0.12
*p*-value	-	0.0169	0.0077	0.2247
72 h	0.17 ± 0.04	0.31 ± 0.02	0.44 ± 0.12	0.26 ± 0.10
*p*-value	-	0.0003	0.0002	0.1018
96 h	0.16 ± 0.03	0.45 ± 0.12	0.69 ± 0.18	0.29 ± 0.08
*p*-value	-	0.0008	0.0002	0.0084
120 h	0.15 ± 0.02	0.51 ± 0.13	0.78 ± 0.21	0.31 ± 0.15
*p*-value	-	0.0004	0.0001	0.0483

**Table 6 ijms-22-09162-t006:** *TIM-3* gene expression in 2 μL/mL GCF-treated CD3+ T cells treated with different concentrations of allicin (Mean ± SD).

Treatment with Allicin	Control w/o Treatment	5 μM	10 μM	25 μM	50 μM
24 h	0.12 ± 0.01	0.26 ± 0.07	0.21 ± 0.02	0.16 ± 0.08	0.15 ± 0.04
*p*-value	-	0.0018	<0.0001	0.3078	0.1321
48 h	0.31 ± 0.01	0.34 ± 0.09	0.29 ± 0.15	0.21 ± 0.02	0.18 ± 0.01
*p*-value	-	0.473	0.7794	<0.0001	<0.0001
72 h	0.38 ± 0.11	0.43 ± 0.11	0.38 ± 0.10	0.26 ± 0.03	0.22 ± 0.02
*p*-value	-	0.4898	>0.9999	0.0408	0.0108
96 h	0.73 ± 0.02	0.79 ± 0.02	0.71 ± 0.18	0.52 ± 0.01	0.45 ± 0.09
*p*-value	-	0.0012	0.8156	<0.0001	0.0001
120 h	0.79 ± 0.22	0.78 ± 0.03	0.73 ± 0.16	0.55 ± 0.04	0.51 ± 0.23
*p*-value	-	0.9225	0.636	0.0443	0.0868

**Table 7 ijms-22-09162-t007:** *LAG-3* gene expression in 2 μL/mL GCF-treated CD3+ T cells treated with different concentrations of allicin (Mean ± SD).

Treatment with Allicin	Control w/o Treatment	5 μM	10 μM	25 μM	50 μM
24 h	0.19 ± 0.005	0.19 ± 0.10	0.19 ± 0.03	0.17 ± 0.03	0.17 ± 0.08
*p*-value	-	>0.9999	>0.9999	0.2317	0.8104
48 h	0.28 ± 0.08	0.27 ± 0.08	0.25 ± 0.11	0.18 ± 0.02	0.17 ± 0.02
*p*-value	-	0.8525	0.6507	0.0317	0.0232
72 h	0.44 ± 0.12	0.44 ± 0.11	0.39 ± 0.10	0.24 ± 0.06	0.21 ± 0.05
*p*-value	-	>0.9999	0.4907	0.0097	0.0035
96 h	0.69 ± 0.18	0.63 ± 0.01	0.57 ± 0.13	0.43 ± 0.11	0.35 ± 0.09
*p*-value	-	0.4885	0.2648	0.0266	0.0059
120 h	0.78 ± 0.21	0.72 ± 0.03	0.66 ± 0.12	0.56 ± 0.02	0.45 ± 0.02
*p*-value	-	0.539	0.2966	0.0455	0.0076

**Table 8 ijms-22-09162-t008:** List of primers used for PCR analysis.

Gene	Forward Primer	Reverse Primer
TIM-3	5′-TCC AAG GAT GCT TAC CAC CAG-3′	5′-AAC ACA AAT ATC CAC ATT GGC-3′
LAG-3	5′-TCA CAG TGA CTC CCA AAT CCT T-3′	5′-GCT CCA CAC AAA GCG TTC TT-3′
GAPDH	5′-ATG GGG AAG GTG AAG GTC G-3′	5′-GGG GTC ATT GAT GGC AAC AAT A-3′

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
