# Peer review of "Allicin May Promote Reversal of T-Cell Dysfunction in Periodontitis via the PD-1 Pathway"

_ijms, 2021, doi:10.3390/ijms22179162_

Round 1

Reviewer 1 Report

This paper described that (1) allicin may bind to PD-1L in silico, (2) GCF induces TNF-α, CCL2, IL-6, IFN-γ, CXCL8, TIM3 and LAG3, and (3) allicin inhibits TIM3 and LAG3 induction by GCF. The relationship between allicin and PD-L1 is interesting, and the antibiotic allicin's contribution to T-cell activation rather than cytotoxicity is also very interesting. However, since (1), (2), and (3) are independent studies, their relevance is speculative, and the Abstract, Results, and Discussion paragraphs contain much speculation, significant revision or many additional experiments are required.

Major point

  1. The probability that allicin binds to PD-1L was shown in silico, but it is unclear whether it actually binds. Perform additional experiments to prove it, otherwise use such as "may" or "guess" in your article to avoid assertions.
  2. In addition, although allicin may bind to PD-L1, there is no evidence that it inhibits the PD-1 pathway. For example, when PD-L1 is added to T cells, TIM3 and LAG3 are increased. Furthermore, when allicin is added, the increase in TIM3 and LAG3 is inhibited. Additional experiments like this should be conducted. Otherwise, the word "inhibit" should not be used.
  3. Determining "T-cell exhaustion" using only two markers (TIM3 and LAG3) is risky so further markers should be considered. The iQueR Human T Cell Exhaustion Kit may be used. If this is not possible, the phrase "T-cell exhaustion" should not be used.
  4. There is no evidence at all that GCF is related to the PD-1 pathway. For example, if GCF is added to T cells in which PD-1 expression is inhibited by siRNA, the expression of TIM3 and LAG3 is inhibited. Additional experiments such as this are needed.
  5. Fig2 has not been utilized in other experiments. Why did you not examine the expression of TNF-α, CCL2, IL-6, IFN-γ, and CXCL8 in the experiments with the addition of allicin (Figs. 5 and 6)?
  6. In studies of T-cell exhaustion and PD-1, CD8 cells are usually used, why did you use CD3 cells? You should provide evidence.

minor point

  1. Many dentists cannot understand GCF, the official name "Gingival crevicular fluid (GCF)" should be mentioned first and then the abbreviation.
  2. Many dentists cannot even understand red-complex bacteria. You should add an explanation of what kind of bacteria are included.
  1. L119-L121; Why does allicin stimulate immunity while activating the PD-1 pathway to suppress T cells?
  2. The placement of Figs. 5 and 6 is reversed.
  3. In Fig. 5 and 6, were GCF and allicin added at the same time? Or did you pre-treat with allicin?

Author Response

Reviewer 1:

This paper described that (1) allicin may bind to PD-1L in silico, (2) GCF induces TNF-α, CCL2, IL-6, IFN-γ, CXCL8, TIM3 and LAG3, and (3) allicin inhibits TIM3 and LAG3 induction by GCF. The relationship between allicin and PD-L1 is interesting, and the antibiotic allicin's contribution to T-cell activation rather than cytotoxicity is also very interesting. However, since (1), (2), and (3) are independent studies, their relevance is speculative, and the Abstract, Results, and Discussion paragraphs contain much speculation, significant revision or many additional experiments are required.

Major point

  1. The probability that allicin binds to PD-1L was shown in silico, but it is unclear whether it actually binds. Perform additional experiments to prove it, otherwise use such as "may" or "guess" in your article to avoid assertions.

Response: respected reviewer, we have rephrased the sentences with a possibility of allicin binding to PD-1 and not as if it is a confirmed finding as per your advise.

  1. In addition, although allicin may bind to PD-L1, there is no evidence that it inhibits the PD-1 pathway. For example, when PD-L1 is added to T cells, TIM3 and LAG3 are increased. Furthermore, when allicin is added, the increase in TIM3 and LAG3 is inhibited. Additional experiments like this should be conducted. Otherwise, the word "inhibit" should not be used.

Response: Changed the title to “Allicin may promote reversal of T-cell dysfunction in periodontitis via the PD-1 pathway”.

  1. Determining "T-cell exhaustion" using only two markers (TIM3 and LAG3) is risky so further markers should be considered. The iQueR Human T Cell Exhaustion Kit may be used. If this is not possible, the phrase "T-cell exhaustion" should not be used.

Response: We attempted to state that the dysfunction in T cells due to periodontitis leads to exhaustion of T cells. However, we agree with you and with these limited set of experiments conducted in this study, we replaced the word ‘exhaustion’ with ‘dysfunction’ wherever necessary in the context of our research.

  1. There is no evidence at all that GCF is related to the PD-1 pathway. For example, if GCF is added to T cells in which PD-1 expression is inhibited by siRNA, the expression of TIM3 and LAG3 is inhibited. Additional experiments such as this are needed.

Response: This is a first time report and we have hypothesized involvement of PD-1 with a sound proof from in silico analysis. Our future research is focused on the exhaustive research on the pathway and signalling molecules with high throughput analysis.

  1. Fig2 has not been utilized in other experiments. Why did you not examine the expression of TNF-α, CCL2, IL-6, IFN-γ, and CXCL8 in the experiments with the addition of allicin (Figs. 5 and 6)?

Response: The purpose of determining and comparing the cytokine levels in the GCF from healthy and peridontitic subjects was to find out whether the key mediators of T cell dysfunction show increased levels in periodontitis. However, this is quantitative assay and not qualitative. Hence, we did not include it in the further experimentation.

  1. In studies of T-cell exhaustion and PD-1, CD8 cells are usually used, why did you use CD3 cells? You should provide evidence

Response: . Respected reviewer, we would like to humbly present to you that CD3+ cells depict a population helper and cytotoxic T cells which are very critical in periodontitis pathogenesis. Morover, the Th response is also largely dependant on the T cells. It has been proven beyond doubt that antibody production by B cells is dependant on T cells responses. Hence we assessed the effect of allicin on the above cells. However as suggested by the reviewer, future experiments will be performed by us on B cells and DC and the data would be published as a separate manuscript

minor point

  1. Many dentists cannot understand GCF, the official name "Gingival crevicular fluid (GCF)" should be mentioned first and then the abbreviation.

Response: respected reviewer, as per your advice, the word GCF has been expanded and mentioned in the abstract section itself to enhance reader understanding

  1. Many dentists cannot even understand red-complex bacteria. You should add an explanation of what kind of bacteria are included.

Response: Respected reviewer, the term red complex bacteria has been removed and replaced with the word putative periodontal pathogens and the names of the bacteria has been mentioned.

  1. L119-L121; Why does allicin stimulate immunity while activating the PD-1 pathway to suppress T cells?

Response: respected reviewer, allicin has been described as a phytochemical with a biphasic response. It has been demonstrated to have both immunostimulatory action and immunoinhibitory action. And we have found allicin inhibiting the PD-1 pathway thereby reducing the levels of TIM-3 and LAG-3

  1. The placement of Figs. 5 and 6 is reversed.

Response: Corrected the placement of figures and tables.

  1. In Fig. 5 and 6, were GCF and allicin added at the same time? Or did you pre-treat with allicin?

Response: GCF and allicin treatment was given at the same time. We have mentioned the same in the materials and methods section.

Reviewer 2:

Minor comments:

  • In line 3 of the Abstract please write GSF in full

Response : respected reviewer, the GCF terminology has been expanded and written as gingival crevicular fluid

  • In the Introduction, please extend for a few lines on the immunostimulatory role of allicin and provide 2-3 references

Response: respected reviewer, the immunostimulatory role of allicin has been elaborated with an addition of 2 references in the introduction section

3)Apart from GSF stimulation did the authors use another non-specific stimulation in order to induce cytokine production and TIM-1, LAG-3 expression? Did allicin act in a similar manner?

Response: respected reviewer, the present study has utilized GCF as the only source of stimulation of the lymphocytes to assess TIM-3 and LAG-3 expression

Major Comment:

  • The authors analyzed the effect of allicin in magnetically isolated CD3+ cells stimulated with GSF. However, PDL-1 is highly expressed in APCs such as B cells and DCs. Were the experiments replicated in whole PBMC cultures?

Response: Respected reviewer, we would like to humbly present to you that CD3+ cells depict a population helper and cytotoxic T cells which are very critical in periodontitis pathogenesis. Morover, the Th response is also largely dependant on the T cells. It has been proven beyond doubt that antibody production by B cells is dependant on T cells responses. Hence we assessed the effect of allicin on the above cells. However as suggested by the reviewer, future experiments will be performed by us on B cells and DC and the data would be published as a separate manuscript.

Reviewer 2 Report

Minor comments:

1) In line 3 of the Abstract please write GSF in full

2) In the Introduction, please extend for a few lines on the immunostimulatory role of allicin and provide 2-3 references

3)Apart from GSF stimulation did the authors use another non-specific stimulation in order to induce cytokine production and TIM-1, LAG-3 expression? Did allicin act in a similar manner?

Major Comment:

1) The authors analyzed the effect of allicin in magnetically isolated CD3+ cells stimulated with GSF. However, PDL-1 is highly expressed in APCs such as B cells and DCs. Were the experiments replicated in whole PBMC cultures?

Author Response

(The authors gave the same response as above.)

Round 2

Reviewer 1 Report

Responding to my comments, the manuscript is improving.

However, some further revisions are needed.

L44: There are two periods.

L65-: Bacterial names are written in italics.

L122-: Refrerence is indicated by [21] and [22], and subsequent Refs should also be adjusted

L129: “sitmulates” is a misspelling of “stimulates”

.

Add the following explanation to your manuscript.

“CD3+ cells depict a population helper and cytotoxic T cells which are very critical in periodontitis pathogenesis. Morover, the Th response is also largely dependant on the T cells. It has been proven beyond doubt that antibody production by B cells is dependant on T cells responses. Hence we assessed the effect of allicin on the above cells.”

Author Response

Reviewer 1:

Responding to my comments, the manuscript is improving.

However, some further revisions are needed.

 L44: There are two periods.

Response: the 2 periods have been addressed and punctuated correctly

L65-: Bacterial names are written in italics.

Response: respected reviewer , we have changed the font of bacterial names to italics

L122-: Refrerence is indicated by [21] and [22], and subsequent Refs should also be adjusted

Response: Respected reviewr, the references 21 and 22 have been numbered and the subsequent references have been adjusted accordingly

L129: “sitmulates” is a misspelling of “stimulates”

Response: respected reviewer , the spelling is stimulates and has been corrected

Add the following explanation to your manuscript.

“CD3+ cells depict a population helper and cytotoxic T cells which are very critical in periodontitis pathogenesis. Morover, the Th response is also largely dependant on the T cells. It has been proven beyond doubt that antibody production by B cells is dependant on T cells responses. Hence we assessed the effect of allicin on the above cells.”

Response: respected reviewer, the above paragraph has been added to the discussion section.
